# Neural Embedding Alignment Reveals Nonlinear Latent Transformations across Brain Regions

**Hanfei Cao**
Fordham University
hcao37@fordham.edu

**Mikio Aoi**
University of California, San Diego
maoi@ucsd.edu

**Stephen Keeley**
Fordham University
skeeley1@fordham.edu

## Abstract

Latent variable models are powerful tools for characterizing high-dimensional neural population activity, and recent work has extended these models to multi-region settings. However, most existing approaches assume linear relationships between populations, limiting their ability to capture the complex, nonlinear mappings that may exist between brain regions. Nonlinear methods, while more flexible, often yield latent spaces whose structure is not uniquely determined, complicating cross-region comparisons.

We introduce a nonlinear variational framework with an alignment objective inspired by stochastic neighbor embedding. Our method enables explicit control over the degree of alignment between latent spaces via a tunable hyperparameter, allowing representations to remain independent or become aligned to facilitate interpretability. We demonstrate the approach on both synthetic data from a four-layer deep neural network and multi-region neural recordings from the mouse visual cortex. Across both settings, the method successfully aligns latent spaces and reveals how manifolds transform across layers or brain regions, providing a flexible tool for probing neural information transformations.

## 1 Introduction

Understanding how information is transformed across different brain regions is a fundamental question in neuroscience. Low-dimensional latent variable models have proven effective in characterizing high-dimensional neural data [1], and recent work has extended these approaches to multi-region settings [2–5]. However, many of these approaches assume either a linear latent structure within each region or linear interactions between regions [6, 3, 2], assumptions that may limit their ability to capture the nonlinear computations involved in inter-area transformations. Some recent approaches extend flexible nonlinear dynamical models to multi-region settings [5, 7], building upon a significant body of work on latent dynamical models in neuroscience [8–13, 7]. Although these dynamical approaches have provided important insights into multi-region neural computation, the choice of the dynamical assumptions varies widely, and the appropriate model may depend on the specific experimental condition or task of an animal. Here, we take a complementary viewpoint: motivated by image-recognition systems, we treat feedforward neural networks as near-instantaneous computations progressing sequentially across brain regions, and aim to identify a static transformation that acts on a latent space from region to region.

In this work, we introduce the Neural Embedding Alignment Tool (NEAT), a nonlinear latent variable model that extends the standard variational autoencoder (VAE) [14] by incorporating a probabilistic similarity-preserving penalty inspired by Stochastic Neighbor Embedding (SNE) [15]. Our approach regularizes the latent spaces of multiple jointly trained VAEs with an objective that encourages the neighborhood distributions of corresponding data points to align across regions. This allows for a set of manifold representations with gradual variations between across brain regions or neural

network layers. This is conceptually related to similarity-based regularization in prior work [16, 17] but differs in that we train VAEs on multiple neural populations simultaneously while introducing an explicit cross-region alignment term. By combining a nonlinear generative model with neighborhood-preserving alignment, our framework both enables direct comparison of latent spaces and supports hypothesis generation about the functional transformations between network layers or brain areas.

## 2  Model

Consider $k$ sets of neural activity, corresponding either to brain regions or to the activations of a trained deep neural network,

$$\mathbf{X}_k = \{\mathbf{x}_k^{(n)}\}_{n=1}^N, \quad k = 1, \ldots, K.$$

Each observation set has an associated set of latent variables,

$$\mathbf{Z}_k = \{\mathbf{z}_k^{(n)}\}_{n=1}^N, \quad k = 1, \ldots, K,$$

where $n$ indexes either the inputs to the neural network, or a given timebin of neural population data. The mapping from latents to observations is given by deep neural network decoders parameterized by $\phi_k$ for each $\mathbf{X}_k$. The model places independent, standard normal priors on each latent point across all $K$ sets:

$$p(\mathbf{Z}_1, \ldots, \mathbf{Z}_K) = \prod_{k=1}^K \prod_{n=1}^N \mathcal{N}\left(\mathbf{z}_k^{(n)}; 0, I\right).$$

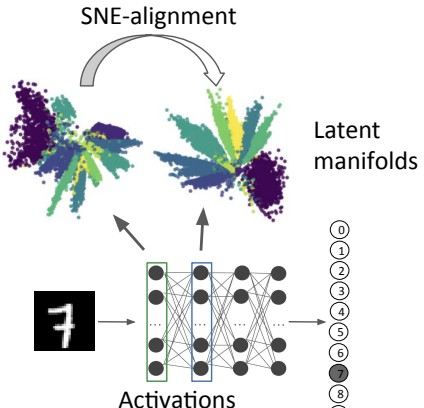

Figure 1: Schematic of our approach in a simulated setting. A fully trained 4 layer deep neural network has activations per layer for each presented image ($\mathbf{X}_k$ where $k$ indicates layer). Each of these activations can be represented with a corresponding latent manifold, which are aligned through a SNE objective in our joint likelihood.

For a latent set $\mathbf{Z} = \{\mathbf{z}^{(n)}\}_{n=1}^N$, we define conditional probabilities $p_{j|i}(\mathbf{Z})$ over pairwise distances in the latent space based on a Gaussian kernel with a global bandwidth $\sigma$:

$$p_{j|i}(\mathbf{Z}) = \frac{\exp\left(-\|\mathbf{z}^{(i)} - \mathbf{z}^{(j)}\|^2/(2\sigma^2)\right)}{\sum_{\ell \neq i} \exp\left(-\|\mathbf{z}^{(i)} - \mathbf{z}^{(\ell)}\|^2/(2\sigma^2)\right)}, \; j \neq i.$$

For each point $i$, this defines a neighborhood distribution $P_i(\mathbf{Z}) = \{p_{j|i}(\mathbf{Z})\}_{j \neq i}$. The full model combines the priors, the likelihood of the observed data (via the decoders), and the pairwise alignment between all latent sets. The unnormalized joint likelihood (for $K$ layers) is:

$$
\begin{aligned}
p(\mathbf{X}_1, \ldots, \mathbf{X}_K, \mathbf{Z}_1, \ldots, \mathbf{Z}_K) \propto & \left[\prod_{k=1}^K p(\mathbf{Z}_k)\right] \prod_{k=1}^K \prod_{n=1}^N p_{\phi_k}\left(\mathbf{x}_k^{(n)} \mid \mathbf{z}_k^{(n)}\right) \\
& \times \exp\left(-\lambda \sum_{1 \leq k < \ell \leq K} \sum_{i=1}^N \mathrm{KL}(P_i(\mathbf{Z}_\ell) \,\|\, P_i(\mathbf{Z}_k))\right)
\end{aligned}
\tag{1}
$$

Here, the latents are aligned in a sequential pairwise manner. That is, layer 1 is aligned with layer 2, layer 2 with layer 3, etc. However, a user can easily change the final term to consider other alignment schemes.

We optimize the model using a standard evidence lower-bound, and our approximate posteriors are independent gaussian distributions whose mean and variance are given via a separate encoding network for each layer $k$, akin to the standard VAE:

$$q_{\theta_k}(\mathbf{z}_k \mid \mathbf{x}_k) \sim \mathcal{N}(\mu_{\theta_k}(\mathbf{x}_k), \sigma_{\theta_k}(\mathbf{x}_k))$$

The variational posterior factorizes across layers and samples:

$$q_\theta(\mathbf{Z}_1, \ldots, \mathbf{Z}_K \mid \mathbf{X}_1, \ldots, \mathbf{X}_K) = \prod_{k=1}^K \prod_{n=1}^N q_{\theta_k}\left(\mathbf{z}_k^{(n)} \mid \mathbf{x}_k^{(n)}\right),$$

The evidence lower bound for this model is:

$$\mathcal{L}_{\text{ELBO}} = \underbrace{\sum_{k=1}^{K} \sum_{n=1}^{N} \mathbb{E}_{q_{\theta_k}}\big[\log p_{\phi_k}(\mathbf{x}_k^{(n)} \mid \mathbf{z}_k^{(n)})\big]}_{\text{Reconstruction Loss}} - \underbrace{\sum_{k=1}^{K} \sum_{n=1}^{N} \text{KL}\Big(q_{\theta_k}(\mathbf{z}_k^{(n)} \mid \mathbf{x}_k^{(n)}) \,\|\, p(\mathbf{z}_k^{(n)})\Big)}_{\text{Prior Regularization}}$$

$$- \lambda \underbrace{\sum_{1 \le k < \ell \le K} \mathbb{E}_{q_{\theta_k}, q_{\theta_\ell}} \left[\sum_{i=1}^{N} \text{KL}\Big(P_i(\mathbf{Z}_\ell) \,\|\, P_i(\mathbf{Z}_k)\Big)\right]}_{\text{Alignment Penalty}} \tag{2}$$

The model is trained by maximizing this objective across all $K$ layers and $N$ data samples (in practice, we train in batches of size $< N$). The first term is the reconstruction loss ensures the latent space accurately reconstructs the input data. This balances with two regularization terms: the second term is the standard VAE prior regularization, ensuring latent representations are compact and the third term is the alignment penalty, which enforces similarity between the probabilistic representation of the latent geometries. Hence, latent geometries will change across layers only when it helps with data reconstruction. We also note that there is a computational cost with increasing the batch size. A large value here is needed for accurate manifold structure estimation in the SNE-inspired penalty, but the last term scales quadratically with increasing the number of samples.

## 3 Results

For each dataset, we first train independent VAEs on each layer or region to estimate the latent dimensionality—successively increasing the dimension until cross-validated predictions explain at least 95% (synthetic) or 70% (real) of the variance in the data. These latent dimensionalities are then fixed for NEAT, where we introduce a penalty term to encourage pairwise alignment between successive latent spaces. By varying the strength of this penalty, we can control the degree of alignment of the latent space across layers or regions. After joint training, we project the latent representations into two dimensions for visualization. To identify functional transformations across layers, we compute the pairwise Euclidean distance between latent points. When two consecutive layers differ in latent dimensionality, we apply CCA [18] to project them into an optimal shared latent space defined by the lower dimensionality of the two latent spaces, and compute the Euclidean distances in this shared subspace. For visualization, we plot the first two principal components of the CCA-reduced latent space, optimally aligned through a procrustes transformation. Finally, we show the mean displacement for latent points corresponding to particular MNIST digits/stimuli across network layers/brain regions.

### 3.1 Synthetic Data

We first trained a four-layer deep neural network (DNN) with 64 nodes per layer on the MNIST dataset for digit classification, achieving a prediction accuracy of 95 %. We extract activations from the trained DNN for each presented digit, and train independent VAEs directly on the four sets of activations to identify dimensionality per layer. With optimal dimensions from each layer fixed at 12, 6, 2 and 2 for layers 1 through 4 respectively, we run our NEAT model for varying alignment strengths ($\lambda$ =0,1,100,1000). To quantify the geometric alignment achieved by NEAT, we calculated the average cosine similarity between the

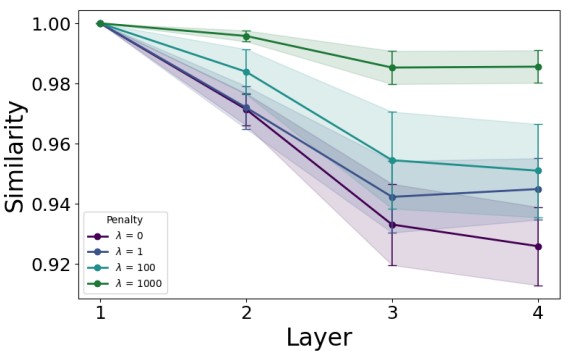

Figure 2: Similarity of latent representations of layers of a deep neural network trained on MNIST for varying alignment penalties ($\lambda = 0, 1, 100, 1000$)

latent representations of all layers after embedding them using the Euclidean embedding scheme with 5 anchor points described by Moschella et al. [19] (Fig 2). As expected in a feedforward network, the

similarity of subsequent layers to the input layer (Layer 1) showed a steady decrease as processing advanced. Importantly, we see an overall increase in the similarity of our latent representations for increasing $\lambda$ values, showing that NEAT successfully aligns latent representations. Furthermore, the reconstruction accuracy of the activations remains very high for all penalty strengths except for $\lambda = 1000$, at which point we observed a decrease in reconstruction fidelity at the expense of maximized latent alignment (see appendix for more details).

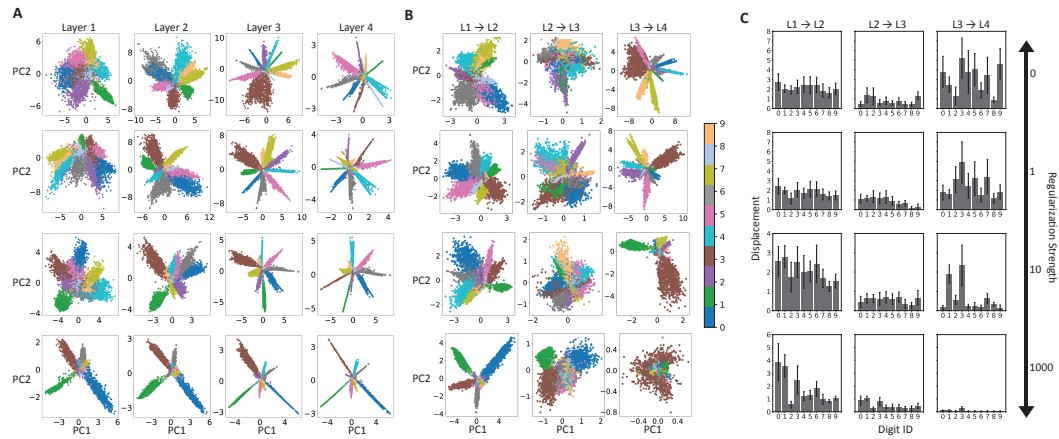

Figure 3: (a) Latent manifold of each NN layer. (b) Visualizing manifold displacement across layers. (c) Magnitude of displacement per digit across layers.

We next visualize information in each DNN layer through two-dimensional projections of the latent spaces (Fig 3A). We see that, irrespective of alignment strength, digits clusters become more segregated as they pass through the neural network, suggesting the network is progressively isolating the digit categories. We note that this can be further quantified by identifying linear classification accuracy of these digit categories in the varying latent spaces (see appendix for more details). In Figure 3B, we visualize the displacement of data points between consecutive layers, highlighting how the representations of specific digits are transformed during layer transitions. For increasing alignment strengths, NEAT identifies displacements that are smaller in magnitude. To further quantify these transitions, we calculate the mean displacement for each digit (Fig 3C). This provides insight into which representations are preferentially transformed between layers. For instance, at a high alignment strength ($\lambda = 100$), digits 1 and 3 exhibit maximal displacement during the Layer 3 to Layer 4 transition. This finding suggests that these final layers may specialize in isolating or refining the features corresponding to these particular digits within this DNN's processing hierarchy.

## 3.2 Visual cortex data

We applied the same framework to mouse neural data recorded from the visual cortex [20], focusing only on time points corresponding to the presentation of four distinct drifting orientation stimuli. Recordings were obtained from the primary visual cortex (V1) and two simultaneously recorded downstream areas: the lateral visual cortex (LM) and the posteromedial visual cortex (PM). Dimensionality of latent spaces was found to be 4 for all regions. A schematic of the visual cortical areas and the visual stimuli (see [20] for details) is shown in Fig. 4A,B. We systematically modulated the degree of alignment between V1 and LM, and between V1 and PM, across three values of $\lambda$ (0, 20, 50).

The resulting manifold transformations illustrate how neural representations evolve as information propagates from V1 to these downstream areas (Fig. 4D). We quantified these changes by measuring the average displacement for each stimulus orientation (Fig. 4E). Notably, at the highest penalty, the transformation from V1 to PM increasingly separates the representations of 90° and 135° gratings from those of 0° and 45°. This finding suggests that the V1-PM interaction serves to refine the categorical distinction between these two groups of stimuli.

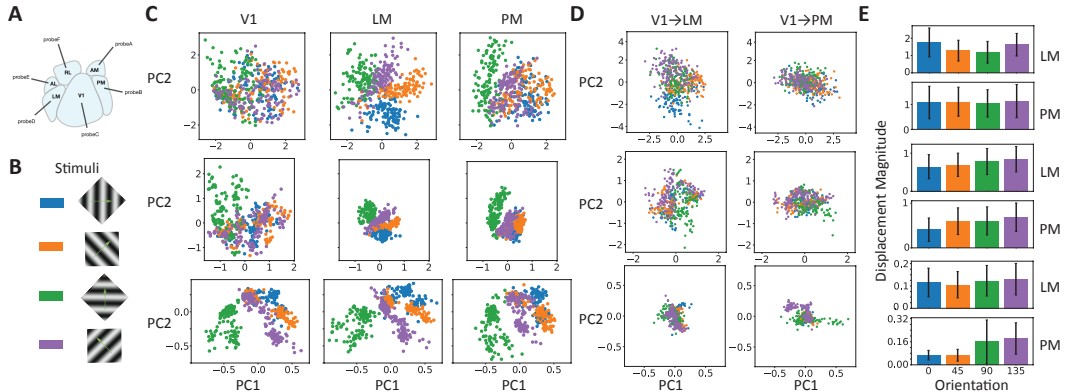

Figure 4: (a) schematic of mouse visual cortical areas and (b) four selected visual stimuli. (c) visualization of the latent spaces (d) visualization of the latent displacements, (e) magnitude of latent displacement categorized by stimuli for three alignment values (top $\lambda = 0$, middle $\lambda = 20$, bottom $\lambda = 50$).

## 4    Conclusion

We introduced NEAT, a tool for quantifying the transformations of latent manifolds in sequential systems like biological and artificial neural networks. Unlike traditional similarity measures of neural network geometries (CKA, SVCCA) [21–24] or techniques that identify a shared subspace for network comparisons [19], NEAT's primary contribution is the ability to use neighborhood structure of a latent representation to inform the latent identification in subsequent layers. This regularization allows a user to find perturbations of latent representations across layers. As such, NEAT provides a principled approach for visualizing nonlinear mappings and generating hypotheses about information processing in both artificial deep networks and in the brain. Future directions include: benchmarking against competing alignment methods, evaluating the effect of different batch sizes, and learning optimal $\lambda$ values.

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
