# Appendix

Table 1: Classification accuracy (%) and variance explained (%) across layers for different $\lambda$ values for our four layer DNN trained on MNIST.

| $\lambda$ | Layer 1 | | Layer 2 | | Layer 3 | | Layer 4 | |
|---|---|---|---|---|---|---|---|---|
| | Acc. | Var. | Acc. | Var. | Acc. | Var. | Acc. | Var. |
| 0 | 0.89 | 0.95 | 0.93 | 0.98 | 0.92 | 0.99 | 0.92 | 0.997 |
| 1 | 0.89 | 0.95 | 0.93 | 0.98 | 0.91 | 0.99 | 0.91 | 0.997 |
| 100 | 0.91 | 0.95 | 0.94 | 0.98 | 0.92 | 0.99 | 0.92 | 0.996 |
| 1000 | 0.91 | 0.93 | 0.93 | 0.97 | 0.92 | 0.98 | 0.92 | 0.993 |

In Table 1, we report layer-wise evaluation across penalty values averaged over 10 fits of NEAT. For each of the four layers, we compute: (i) the test accuracy from a cross-validated linear classifier on the latent representations, indicating how well the latent space separates the ten handwritten digits, and (ii) the variance explained by those latent representations, showing how much of the meaningful structure in the original data is captured by the latent space. The identified latent dimensions were 12, 6, 2 and 2 for layer 1, 2, 3 and 4, respectively. Classification for digits was lower in layer 1 than subsequent layers, demonstrating classification improvement as the information progresses through the network. Furthermore, the introduction of the alignment penalty did not significantly degrade the ability of the VAE to reconstruct the activations until the largest penalty value, suggesting NEAT robustly defines aligned manifolds without sacrificing predictive power.