# OpenReview forum: "Neural Embedding Alignment Reveals Nonlinear Latent Transformations across Brain Regions"
_NeurIPS.cc/2025/Workshop/UniReps — UniReps2025 oral_

### Official Review · Reviewer_LeBS · 2025-09-04
**A simple latent variable method for dissecting representation transformations across layers or brain areas**

**Confidence:** 3

**Review:**

Understanding how information is transformed across layers of a neural network or across areas of the mammalian brain represents a fundamental scientific question. Although some tools based on latent representations are becoming available, there remains a strong need for flexible and simple models. This manuscript contributes by introducing a new, potentially non-linear latent variable method that operates on static data (e.g., instantaneous or time-averaged stimulus responses). The model enables the visualization of both private and shared representational structures across layers via a tunable hyperparameter.
Overall, the work is interesting and worth pursuing. However, it feels preliminary and would benefit further development and applications.

Pros:
* The work is interesting and relevant for the workshop audience.
* The writing is generally clear, though some sections would benefit from additional explanation -- particularly regarding the model’s details and the rationale behind specific modeling choices.

Cons:
* Benchmarking against other approaches is currently missing.
* The data experiments and analyses could be more in-depth. In particular, I do not fully understand why the authors limited the visualization of the latents to only two components. It would have been interesting to know how many latent dimensions were identified in each layer/area, what their structure looked like, and how this number varies as a function of the hyperparameter λ.
* The typical data analysis pipeline envisioned with this method is unclear. What type of insights do the authors hope to achieve? The ability to vary λ makes the framework flexible, but how should λ be used in practice? Should one scan across values of λ to probe private and shared variability across layers/areas? And how does this analysis provide insight into the non-linear transformations applied to shared latents by each layer/area?

**Score:**

3

**Topic Fit:**

2

---

### Official Review · Reviewer_tdya · 2025-09-15
**Promising framework with strong evaluations, but limited by scalabiity and missing baselines.**

**Confidence:** 4

**Review:**

**Summary:** Paper introduces a VAE-based framework with an SNE-inspired tunable alignment loss to compare latent spaces across brain regions or neural network layers.


**Strengths**
* Idea is clearly presented and evaluated using both quantitative and qualitative analyses.
* Introduces a tunable alignment architecture across latent spaces, which looks like a promising direction, although the practical importance could be emphasized more clearly.
* Demonstrates results on both synthetic and real neural data.
* The model is optimized over different latent dimensions, with accompanying details and explained variance reported.


**Weaknesses/Questions**
* **Scalability** The alignment cost scales quadratically with batch size, which may limit applications to larger datasets. As mentioned in the conclusion, further analysis of batch size effects would clarify these limitations.
* **Benchmarks** The lack of comparison with the previous methods makes it difficult to assess the novelty of the work.
* **Interpretability** Interpretability of the alignment hyperparamater is not explored, and a clearer justification of its chosen values would improve the interpretability.
* **Hyperparameter tuning** More extensive tuning and ablation studies are needed.
* **Implementation details** Key details for reproducibility such as training epochs, optimization method and batch sizes are missing.

**Score:**

4

**Topic Fit:**

3

---

### Official Review · Reviewer_hecE · 2025-09-15
**VAE-based alignment method with clear potential, but interpretability and missing baselines limit conclusions**

**Confidence:** 3

**Review:**

Summary
This work introduces NEAT, a variational autoencoder framework augmented with a stochastic neighbor embedding (SNE) penalty to encourage alignment between latent spaces of neural populations or neural network layers. The goal is to capture nonlinear manifold transformations across regions. The method is evaluated on two settings: MNIST-trained neural network activations and multi-region mouse visual cortex data, showing that the approach can enforce varying degrees of alignment and visualize representational transformations.

Strengths
- Addresses a timely and important problem: probing how representations evolve across layers or brain regions.
- Builds on the well-understood and flexible VAE framework, making the method accessible and extensible.
- Introduces a tunable alignment hyperparameter, which is an intuitive and useful way to study the continuum between independence and alignment.
- Demonstrates applications both on synthetic (DNN layers) and biological (mouse visual cortex) data, supporting generality.
- Clear potential to stimulate discussion at the workshop about representational geometry and cross-region comparisons.

Weaknesses / Limitations
- The reconstruction loss is not reported; it is important to quantify the trade-off between generative fidelity and alignment. Without this, it is hard to interpret the quality of the learned representations.
- The method enforces alignment rather than probing naturally occurring structure. This raises the question: what do we actually learn beyond the fact that we can make representations look similar? For both datasets, it remains somewhat unclear what biological or computational insights emerge.
- Some technical aspects are underexplained. For instance, the meaning of “sequential pairwise alignment” (Eq. 1) should be clarified, as well as the interpretation of “explaining 95% / 70% of the data” when determining latent dimensionality.
- Related work is incomplete. There are relevant contributions on e.g. relative anchor point embeddings (Moschella 2022) which should be acknowledged.
- Visualizations (e.g. Fig. 2) could be made more interpretable. Flow fields or displacement vectors might convey transformations more clearly than only plotting points. (just as a suggestion)
- Some observed artifacts (reflections in Fig. 2A, layer 4, 2nd and 3rd row) are not discussed: are they due to PCA or alignment procedure?

Questions / Suggestions for Authors
- Out of interest: How does this approach relate to hierarchical VAEs, which also impose structure across multiple latent levels?
- Could the framework be extended to probe whether some alignments emerge naturally (e.g. at specific λ values) rather than always enforcing them?
- Could this approach be benchmarked against existing similarity-based methods (e.g., CCA, CKA) to highlight added value?

Overall Assessment
This is a promising contribution with clear relevance to UniReps. While technically preliminary, it introduces a novel perspective and has the potential to spark useful discussion about nonlinear representational alignment in both neuroscience and machine learning. The main limitation is interpretability: it remains somewhat unclear what scientific conclusions follow from enforcing alignment. Nevertheless, the clarity of the proposal, the link to existing VAE methods, and the potential for discussion justify acceptance as an Extended Abstract.


Moschella, L., Maiorca, V., Fumero, M., Norelli, A., Locatello, F., & Rodolà, E. (2022, September 29). Relative representations enable zero-shot latent space communication. The Eleventh International Conference on Learning Representations. https://openreview.net/forum?id=SrC-nwieGJ

**Score:**

4

**Topic Fit:**

3

---

### Official Review · Reviewer_Qpdg · 2025-09-15

**Confidence:** 5

**Review:**

### Summary

This paper introduces NEAT, a nonlinear VAE framework with an SNE-inspired alignment objective for comparing latent spaces across brain regions or neural network layers. The method provides a tunable hyperparameter that controls alignment strength, enabling exploration of representational transformations. The approach is clearly presented and validated on both synthetic and biological data. While technically sound and potentially impactful, the interpretability claims — especially the use of “mean displacement” — require stronger justification, and the motivation for why a practitioner should prefer NEAT over simpler representational alignment methods could be more clearly articulated.

### Review

The paper is well-motivated and introduces NEAT, a nonlinear VAE framework with a neighborhood-preserving alignment penalty. I found the method technically solid, and the experiments on both synthetic DNN activations and mouse visual cortex recordings illustrate its versatility. The idea of tuning alignment between latent spaces is novel and potentially impactful for understanding representational transformations.

That said, I have concerns about **interpretability claims**, especially in the synthetic example. The paper interprets the **mean displacement** of latent representations (e.g., digits 1 and 3 moving more between layer 3→4) as evidence that those digits are isolated by the final layer. However, it is unclear why displacement — defined as the Euclidean shift of aligned embeddings — should be taken as a meaningful measure of functional specialization. A large shift could reflect many factors (variance scaling, embedding geometry) and does not necessarily indicate interpretability or task relevance.

**Pros**
- Clear and novel modeling contribution (multi-region VAE with tunable alignment).
- Strong motivation: bridging nonlinear flexibility with latent comparability.
- Dual validation on artificial and biological datasets.
- Elegant visualization of manifold transformations.

**Cons**
- Justification of **displacement as an interpretability metric** is weak; alternative similarity measures (RSA, SVCCA, CCA-based approaches) could provide more robust benchmarks.
- Motivation for why a practitioner should apply NEAT over simpler representational alignment tools is underspecified.
- Limited biological interpretation of the mouse cortex example.
- Scalability issues (alignment penalty scales quadratically with batch size) are acknowledged but not deeply explored.

**Score:**

4

**Topic Fit:**

3